# Predicting Maximum and Cumulative Response of A Base-isolated Building Using Pushover Analysis

**Kenji Fujii** [1],***, Yoshiyuki Mogi** [1] **and Takumi Noguchi** [1]

Department of Architecture, Faculty of Creative Engineering, Chiba Institute of Technology, Chiba 275-0016, Japan; ym.takkyu0921drive@gmail.com (Y.M.); ntakumi427@gmail.com (T.N.)
* Correspondence: kenji.fujii@it-chiba.ac.jp

**Abstract:** The evaluation of the maximum and cumulative response is an important issue for the seismic design of new base-isolated buildings. This study predicts the maximum and cumulative response of a 14-story reinforced concrete base-isolated building using a set of pushover analyses. In the proposed pushover analysis method, the maximum and cumulative responses of the first and higher modes are evaluated from the nonlinear analysis of equivalent single-degree-of-freedom (SDOF) models. Then, the maximum local responses are predicted by enveloping the two pushover analysis results by referring to the contribution of the first and higher modal responses, while the cumulative strain energies of the lead-rubber bearings and steel dampers are predicted from the cumulative response of the first mode. The results reveal that the responses predicted by the proposed set of pushover analyses have satisfactory accuracy.

**Keywords:** base-isolated building; pushover analysis; higher-mode effect; cumulative response

## 1. Introduction

Seismic isolation is one of the most attractive techniques for the earthquake protection of buildings. The motivation for applying seismic isolation is to reduce the displacement response of structures, which is directly related to the damage of structural and nonstructural elements, and it is to reduce the acceleration response of floors related to nonstructural elements [1–5]. In the case of a properly designed seismic isolated building, most gravity-load-carrying members remain elastic and minor damage to nonstructural elements is expected. The reason for this is that seismic isolation can reduce both the deformation and acceleration, and most of the seismic energy input is absorbed into the isolated layer. Therefore, seismic isolated buildings remain operational after extreme earthquakes. Moreover, as described in [1], the complicated structural details required for traditional earthquake-resistant structures, such as a large amount of shear reinforcement at the expected hinge zone of the beam-end, may be simplified for the supported superstructures of properly designed base-isolated buildings.

For base-isolated buildings, several studies on the nonlinear seismic responses have considered the nonlinear behavior of superstructures [6–8]. These studies have concluded that the positive aspect of superstructure nonlinearity, such as the elongation of the natural period and increase of energy absorption, should not be expected in the case of isolated structural yielding. Therefore, the design guidelines assume an elastic behavior for the superstructure of base isolated buildings [9]. Thus, the maximum force demand of the superstructure's members must be properly evaluated.

The evaluation of the floor acceleration and cumulative strain energy absorbed in the isolated layer are also important issues for the proper seismic design of base-isolated buildings. Calvi and Ruggiero [10] have pointed out that the contribution of the higher mode may be significant for the floor acceleration of base-isolated buildings, and negligible with regard to floor displacement. According to the behavior of isolators and dampers under cyclic loading, the Architectural Institute of Japan

(AIJ) has investigated the structural response under long-period seismic ground motions [11], and has strongly emphasized the importance of evaluating the cumulative energy, and not only the maximum displacement, for long period ground motions. Recent studies provided several tests for lead rubber bearings (LRBs), high damping rubber bearings (HDBs) [12], and steel dampers [13]. These studies highlight the importance of the cumulative response at the isolated layer. Specifically, the test results presented in [12] reveal that the yield strength of LRB specimens dropped to approximately 50% under a sine wave input that far exceeded the energy of actual seismic motions, owing to the increase in the lead plug temperature.

The seismic design recommendations of the seismic isolated buildings published from AIJ [11] emphasize the importance of the proper confirmations about the maximum displacement and axial force of isolators, and the required energy absorptions of dampers. For this purpose, there is no doubt that the most rigorous way to get the maximum and cumulative response is the nonlinear time-history analysis. However, it requires large analysis loads due to the preparing of more detailed structural and ground motion data. Besides, the interpretation and understanding of the analysis results may require sizable knowledge and the experiences of designers and structural analysts. Therefore, there are several studies that agree with the application of a simplified nonlinear analysis procedure to base-isolated buildings [14–21]. The simplified nonlinear analysis procedure, which combines the nonlinear static (pushover) analysis of a multi-degree-of freedom (MDOF) model with the response spectrum analysis of an equivalent single-degree-of-freedom (SDOF) model [22–25], is widely applied to the traditional earthquake-resistant structures. However, to the authors' knowledge, there are few studies on (a) the demand of the member forces in the superstructure, (b) the maximum floor accelerations, and (c) the cumulative energies absorbed in the isolated layer of base-isolated buildings.

The authors proposed a procedure to predict the force demand in the superstructure's members of base-isolated buildings by means of the pushover analysis in a recent paper [26]. Such a procedure was based on estimating the contribution of the first and higher modes (second) to the peak response. Therefore, it is essential to predict these modal responses with accuracy for the better prediction of the force demand and other engineering parameters.

This study predicted the maximum and cumulative seismic response of a 14-story reinforced concrete base-isolated frame building using pushover analysis and nonlinear time-history analysis for the equivalent SDOF models. In this study, the peak of the nonlinear first and second modal responses are predicted from the time-history analysis of two independent equivalent SDOF models. Specifically, the maximum equivalent acceleration of the second mode is predicted considering the change of the second effective modal mass due to the nonlinearity of the isolation layer. For simplicity, nonlinear behavior was only assumed in the isolated layer, while the behavior of the superstructure was assumed as linearly elastic. The isolated layer of the building was comprised of natural rubber bearings (NRBs), LRBs, and steel dampers. The target engineering parameters predicted in this study are the following: (i) maximum relative floor displacement, (ii) maximum absolute floor acceleration, (iii) maximum shear forces of vertical members in the superstructure, (iv) maximum shear strain and nominal stress of isolator, and (v) cumulative strain energies of LRBs and dampers.

The rest of the paper is organized as follows. Section 2 describes the concept of and outlines the proposed procedure for predicting the maximum and cumulative response of base-isolated buildings. Section 3 provides basic information about the model building and the ground motions. Section 4 describes the nonlinear time-history analysis of the base-isolated frame building and presents a comparison with the predicted maximum and cumulative response obtained though the proposed procedure. Section 5 discusses the accuracy of the predicted maximum and cumulative modal response by calculating the nonlinear modal responses of the first and second modes using a method proposed in a previous paper by the authors [26].

## 2. Description of the Proposed Procedure

### 2.1. Basis Procedure

The following assumptions are made:

1. The considered base-isolated building predominantly oscillates in the first mode.
2. The superstructure behavior is linearly elastic, while nonlinear behavior is assumed only in the isolated layer.
3. The local response (e.g., floor displacement and acceleration) can be approximated by a combination of the first and second modal responses.
4. The cumulative response in the isolated layer (e.g., the cumulative strain energy of the damper) can be approximated by the first modal contribution.

Figure 1 shows the basic concepts of the two equivalent SDOF models representing the first and second modal responses used in the proposed procedure.

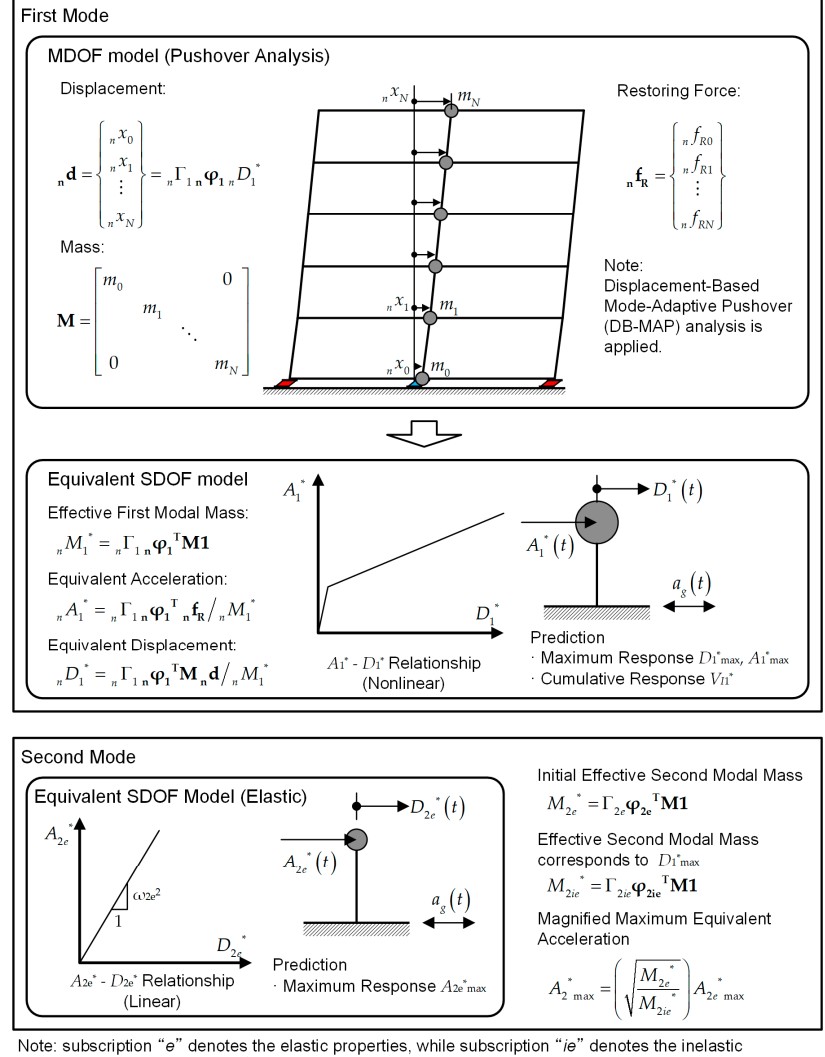

**Figure 1.** Concept of two equivalent single-degree-of-freedom (SDOF) models representing the first and second modal responses used in the proposed procedure.

In the proposed procedure, the behavior of the equivalent SDOF model representing the first modal response is assumed as nonlinear, while the behavior of the equivalent SDOF model representing

the second modal response is assumed as linearly elastic. These assumptions are the same as those made in a previous study [25], wherein the higher mode response was approximated by assuming linearly elastic behavior.

*2.2. Outline of the Procedure*

2.2.1. Step 1: Pushover Analysis of the Building Model (First Mode)

The pushover analysis of the *N*-story base-isolated frame building model is carried out by considering the change in the first mode's shape at each nonlinear stage. Displacement-based mode-adaptive pushover (DB-MAP) analysis is applied [27,28]. Then, the nonlinear equivalent-acceleration-equivalent-displacement ($A_1^*$-$D_1^*$) relationship of the equivalent SDOF model is determined based on the pushover analysis results. The equivalent displacement and acceleration at step *n* (namely, $_nD_1^*$ and $_nA_1^*$) are determined using Equations (1) and (2), respectively, assuming that the displacement vector $_n\mathbf{d}$ is proportional to the first mode vector $_n\Gamma_{1\mathbf{n}}\boldsymbol{\varphi}_1$ at each loading step as follows:

$$_nD_1^* = \frac{_n\Gamma_{1\mathbf{n}}\boldsymbol{\varphi}_1{}^{\mathbf{T}}\mathbf{M}_{\mathbf{n}}\mathbf{d}}{_nM_1^*} = \left(\sum_{j=0}^{N} m_j\,_nx_j^2\right)\Big/\left(\sum_{j=0}^{N} m_j\,_nx_j\right),\tag{1}$$

$$_nA_1^* = \frac{_n\Gamma_{1\mathbf{n}}\boldsymbol{\varphi}_1{}^{\mathbf{T}}\mathbf{M}_{\mathbf{n}}\mathbf{f}_{\mathbf{R}}}{_nM_1^*} = \left(\sum_{j=0}^{N} {}_nf_{Rj}\,_nx_j\right)\Big/\left(\sum_{j=0}^{N} m_j\,_nx_j\right),\tag{2}$$

$$_nM_1^* = {}_n\Gamma_{1\mathbf{n}}\boldsymbol{\varphi}_1{}^{\mathbf{T}}\mathbf{M1} = \left(\sum_{j=0}^{N} m_j\,_nx_j\right)^2\Big/\left(\sum_{j=0}^{N} m_j\,_nx_j{}^2\right),\tag{3}$$

$$_n\Gamma_1 = \frac{_{\mathbf{n}}\boldsymbol{\varphi}_1{}^{\mathbf{T}}\mathbf{M1}}{_{\mathbf{n}}\boldsymbol{\varphi}_1{}^{\mathbf{T}}\mathbf{M}_{\mathbf{n}}\boldsymbol{\varphi}_1} = \left(\sum_{j=0}^{N} m_j\,_nx_j\right)\Big/\left(\sum_{j=0}^{N} m_j\,_nx_j{}^2\right).\tag{4}$$

In Equations (1)–(4), $m_j$ is the mass of the *j*th floor; $_nM_1^*$ is the effective first modal mass at step *n*. The $A_1^*$-$D_1^*$ relationship obtained from the pushover analysis result is idealized as a bilinear curve, and the normal bilinear hysteresis rule is adopted to model the nonlinear behavior of the first mode. Viscous damping is not considered for the equivalent SDOF model, because the energy absorption of the first mode is already included in the hysteresis energy.

2.2.2. Step 2: Nonlinear Analysis of the Equivalent SDOF Model (First Mode)

A nonlinear time-history analysis of the equivalent SDOF model is carried out to obtain the maximum response (maximum equivalent displacement $D_1^*{}_{max}$ and maximum equivalent acceleration $A_1^*{}_{max}$) and the cumulative response (the equivalent velocity of the cumulative input energy [29] is calculated for the first modal response) as follows:

$$V_{I1}^* = \sqrt{2\int_0^{t_d} e_{I1}^*(t)dt},\, e_{I1}^*(t) = -\dot{D}_1^*(t)a_g(t).\tag{5}$$

In Equation (5), $e_{I1}^*(t)$ is the energy rate of the first modal response per unit mass, $\dot{D}_1^*(t)$ is the equivalent velocity of the first modal response, and $a_g(t)$ is the ground acceleration, defined within the range $[0, t_d]$.

### 2.2.3. Step 3: Prediction of Local Cumulative Response from First Mode

The loading step corresponding to $D_1^*{}_{max}$ is determined from the results of step 1. Then, the first mode vector and the effective modal mass of the first mode corresponding to $D_1^*{}_{max}$, namely, $\Gamma_{1ie}\boldsymbol{\varphi_{1ie}}$ and $M_{1ie}^*$, are determined, respectively, and the maximum horizontal deformation of the isolators and dampers at the isolated layer $\delta_{Dk}$ (= $x_{01max}$: the relative displacement at the 0th floor corresponding to $D_1^*{}_{max}$) is also determined. The cumulative strain energy of the $k$th damper, $E_{SDk}$, is calculated as follows:

$$E_{SDk} = \begin{cases} \dfrac{e_{SDk}}{\sum\limits_{k} e_{SDk}} \cdot \dfrac{1}{2}M_{1ie}^* V_{I1}^{*2} & : \sum\limits_{k} e_{SDk} > 0 \\ 0 & : \sum\limits_{k} e_{SDk} = 0 \end{cases} \tag{6}$$

$$e_{SDk} = \begin{cases} Q_{yDk}\left(\delta_{Dk} - \delta_{yDk}\right) & : \delta_{Dk} \geq \delta_{yDk} \\ 0 & : \delta_{Dk} < \delta_{yDk} \end{cases} . \tag{7}$$

In Equation (6), $e_{SDk}$ is the plastic strain energy of the $k$th damper under monotonic loading calculated from the pushover analysis results. While in Equation (7), $Q_{yDk}$ and $\delta_{yDk}$ denote the yield strength and displacement of the $k$th damper, respectively.

### 2.2.4. Step 4: Calculation of Equivalent SDOF Model Properties (Second Mode)

The second mode vector $\Gamma_{2ie}\boldsymbol{\varphi_{2ie}}$ is calculated from Equations (8) and (9) in terms of $\Gamma_{1ie}\boldsymbol{\varphi_{1ie}}$, and the second mode vector in the elastic range ($\Gamma_{2e}\boldsymbol{\varphi_{2e}}$). The vector $\boldsymbol{\varphi_{2ie}}$ is determined to satisfy the orthogonality conditions of the two mode vectors, $\boldsymbol{\varphi_{1ie}}$ and $\boldsymbol{\varphi_{2ie}}$:

$$\boldsymbol{\varphi_{2ie}} = \boldsymbol{\varphi_{2e}} - \frac{\boldsymbol{\varphi_{2e}}^{\mathbf{T}}\mathbf{M}\boldsymbol{\varphi_{1ie}}}{\boldsymbol{\varphi_{1ie}}^{\mathbf{T}}\mathbf{M}\boldsymbol{\varphi_{1ie}}}\boldsymbol{\varphi_{1ie}} \tag{8}$$

$$\Gamma_{2ie} = \boldsymbol{\varphi_{2ie}}^{\mathbf{T}}\mathbf{M1} / \boldsymbol{\varphi_{2ie}}^{\mathbf{T}}\mathbf{M}\boldsymbol{\varphi_{2ie}} \tag{9}$$

Then, the effective second modal mass is calculated in terms of $\Gamma_{2ie}\boldsymbol{\varphi_{2ie}}$ as follows:

$$M_{2ie}^* = \Gamma_{2ie}\boldsymbol{\varphi_{2ie}}^{\mathbf{T}}\mathbf{M1} \tag{10}$$

Next, the equivalent SDOF model properties are determined. The natural period of the equivalent elastic SDOF model is set as the initial (elastic) natural period of the second mode of the base-isolated frame building model, $T_{2e}$. The viscous damping ratio of the elastic SDOF model, $h_{2e}$, is calculated as follows:

$$h_{2e} = h_{1fix} \cdot \left(T_{1fix} / T_{2e}\right) \tag{11}$$

where $T_{1fix}$ and $h_{1fix}$ are the natural period and viscous damping ratio of the first mode of the non-base-isolated frame building model, respectively. In this study, the damping ratio $h_{2e}$ is proportional to the ratio of the natural frequency of the natural modes because the damping of the superstructure is assumed to be proportional to the stiffness matrix of the superstructure, as described later.

### 2.2.5. Step 5: Linear Analysis of Equivalent SDOF Model (Second Mode)

The time-history analysis of the linear equivalent SDOF model is carried out to obtain the maximum response (maximum equivalent acceleration $A_{2e}^*{}_{max}$). Then, the maximum equivalent acceleration of the second mode is calculated by taking into account the change in the effective modal mass as follows:

$$A_2^*{}_{max} = \left(\sqrt{\frac{M_{2e}^*}{M_{2ie}^*}}\right)A_{2e}^*{}_{max} \tag{12}$$

In Equation (12), $M_{2e}^*$ is the effective second modal mass in the elastic range. The derivation of Equation (12) is discussed in the Appendix A.

2.2.6. Step 6: Prediction of Local Maximum Response Considering the Contribution of the Second Mode

This step involves the procedure proposed by the authors in a previous paper [26]. First, the set of the horizontal-force distributions $\mathbf{p^+}$ and $\mathbf{p^-}$ is calculated in terms of $\Gamma_{1ie}\boldsymbol{\varphi}_{\mathbf{1ie}}$ and $\Gamma_{2ie}\boldsymbol{\varphi}_{\mathbf{2ie}}$ as follows:

$$\begin{cases} \mathbf{p^+} = \mathbf{M}(\Gamma_{1ie}\boldsymbol{\varphi}_{\mathbf{1ie}}A_1{}^*{}_{\max} + 0.5\Gamma_{2ie}\boldsymbol{\varphi}_{\mathbf{2ie}}A_2{}^*{}_{\max}) \\ \mathbf{p^-} = \mathbf{M}(\Gamma_{1ie}\boldsymbol{\varphi}_{\mathbf{1ie}}A_1{}^*{}_{\max} - 0.5\Gamma_{2ie}\boldsymbol{\varphi}_{\mathbf{2ie}}A_2{}^*{}_{\max}) \end{cases} \tag{13}$$

Figure 2 shows the concept of the horizontal force distribution obtained from the combination of the two modal responses.

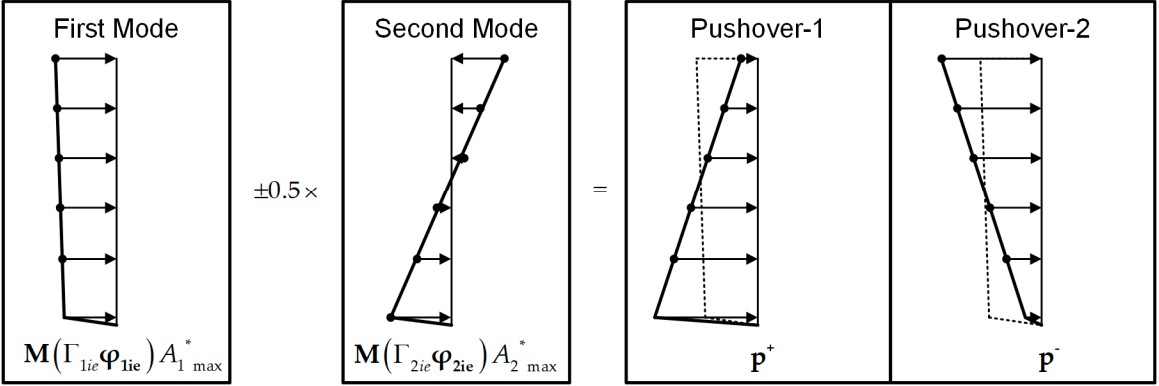

**Figure 2.** Concept of horizontal force distribution obtained from combination of two modal responses.

Then, a pushover analysis is performed using the invariant force distributions $\mathbf{p^+}$ and $\mathbf{p^-}$ (referred to as Pushovers 1 and 2, respectively) until the equivalent displacement $_nD^*$ reaches $D_1{}^*{}_{\max}$, using the following calculation formula:

$$_nD^* = \frac{\Gamma_{1ie}\boldsymbol{\varphi}_{\mathbf{1ie}}{}^{\mathbf{T}}\mathbf{M_n}\mathbf{d}}{M_{1ie}{}^*} \tag{14}$$

Next, the maximum response from the envelope of Pushovers 1 and 2 is determined, that is, the maximum relative displacement, maximum restoring force, and maximum member forces are obtained as the maximum value of the Pushovers 1 and 2 results. The maximum floor acceleration $\mathbf{a_{max}}$ is calculated as follows:

$$\mathbf{a_{max}} = \mathbf{M^{-1}}\mathbf{f_{Rmax}} \tag{15}$$

where $\mathbf{f_{Rmax}}$ is the maximum restoring force vector obtained from the Pushovers 1 and 2 envelopes.

## 3. Building and Ground Motion Data

### 3.1. Building Data

This study investigated the base-isolated building model shown in Figure 3. This model is a 14-story base-isolated reinforced concrete building model, which is a variant from that used in the previous study [26]. Spandrel walls are added at column $X_2Y_2$ in the X-direction and columns $X_1Y_2$ and $X_3Y_2$ in the Y-direction. This study only considered horizontal earthquake excitation in the X-direction. Figure 4 shows some details of column $X_2Y_2$ with the spandrel wall. The thickness of the spandrel wall is assumed as 150 mm. The floor weights per unit area above level $Z_1$ and at level $Z_0$ are assumed as 14 and 32 kN/m$^2$, respectively. The floor masses above level $Z_1$ and at level $Z_0$ are 504.9 ton and 1154 ton, respectively. Therefore, the total mass above the isolated layer $M$ is 8223 ton (total weight $W$ = 80581 kN). The behavior of all beams and columns is assumed to be linearly elastic.

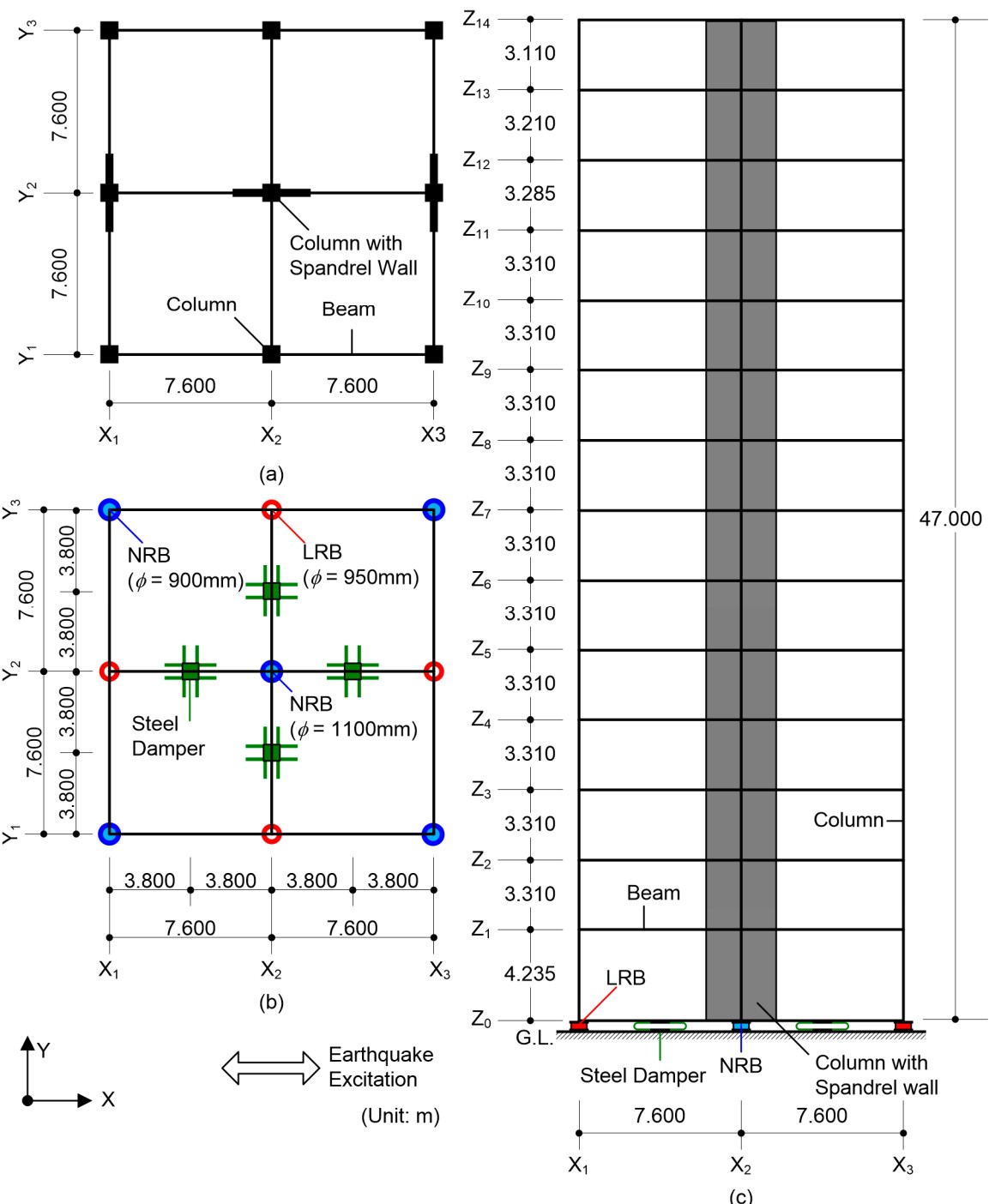

**Figure 3.** Simplified structural plan and elevation of the base-isolated building considered in this study: (**a**) plan of levels $Z_1$ to $Z_{14}$; (**b**) plan of level $Z_0$; (**c**) elevation of frame $Y_2$.

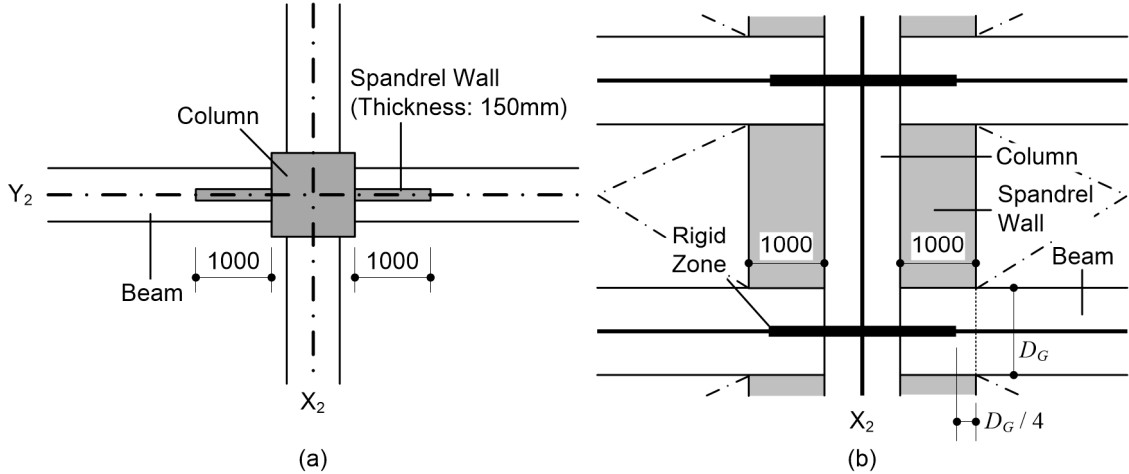

**Figure 4.** Details of column $X_2Y_2$ with spandrel wall: (**a**) plan view; (**b**) elevation view.

The properties of the isolated layer are the same as those in the model used in a previous study by the authors [26]. Figure 5 shows the force-deformation relationships for the isolators and damper. The behavior of the NRB is assumed as linearly elastic while that of the LRB and the behavior of the steel damper are assumed as bilinear.

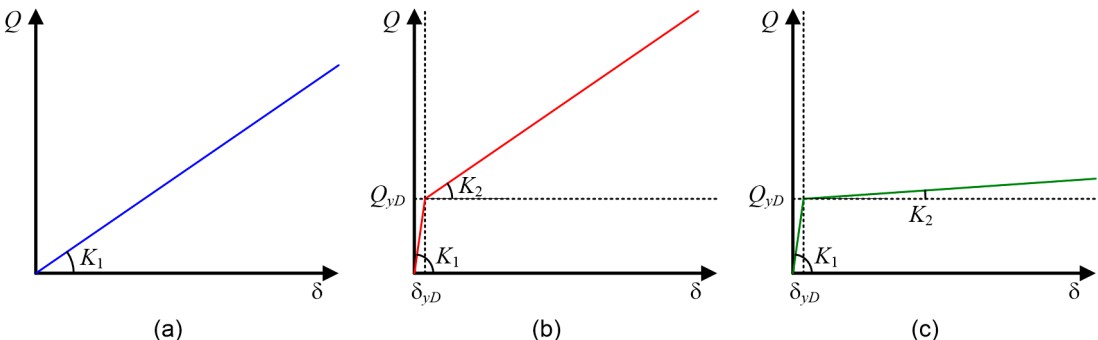

**Figure 5.** Force-deformation relationships for isolators and damper: (**a**) NRB, (**b**) LRB, and (**c**) steel damper.

The two parameters of the isolated layer, the isolated period $T_f$ and the shear force coefficient of dampers $\alpha_s$, are defined, respectively, as follows:

$$T_f = 2\pi \sqrt{M/K_f} \tag{16}$$

$$\alpha_s = {}_sQ_y/W = \sum_k Q_{yDk}/W \tag{17}$$

In Equation (16), $K_f$ is the total stiffness of the isolated layer without dampers (calculated as the sum of the initial stiffness $K_1$ of the NRBs and the post yielding stiffness $K_2$ of the LRBs). In Equation (17), ${}_sQ_y$ is the sum of the yield strength of the damper $Q_{yDk}$ (including the LRBs). The calculated values of $T_f$ and $\alpha_s$ are $T_f = 4.84$ s and $\alpha_s = 0.048$. The modelling scheme for the frame building is the same as that in the previous study by the authors [26].

Figure 6 shows the shape of the first and second mode vectors of the non-isolated and isolated building models in the elastic range (X-direction). Notably, the mode shapes shown in Figure 6a, referred to as non-base-isolated, are the vectors obtained under the assumption that the superstructure is pin-supported. Here, $T_{ke}$ is the $k$th natural period in the elastic range ($k = 1,2$) and $m_{ke}^*$ is the effective modal mass ratio of the $k$th mode.

As shown in Figure 6, the equivalent first modal mass ratio in the non-base-isolated case is 0.758, while the modal mass ratio of the base-isolated case is 0.971. The difference of the two cases is attributed to the difference of the first mode shapes; the first mode shape of the non-base-isolated case is similar to an inverted triangular shape, while that of the base-isolated case exhibited significant deformation in the isolated layer and the superstructure behaved as a rigid body. The isolated period of the building model is $T_f$ = 4.84 s and the ratio $T_f/T_{1fix}$ = 4.84/0.854 = 5.67.

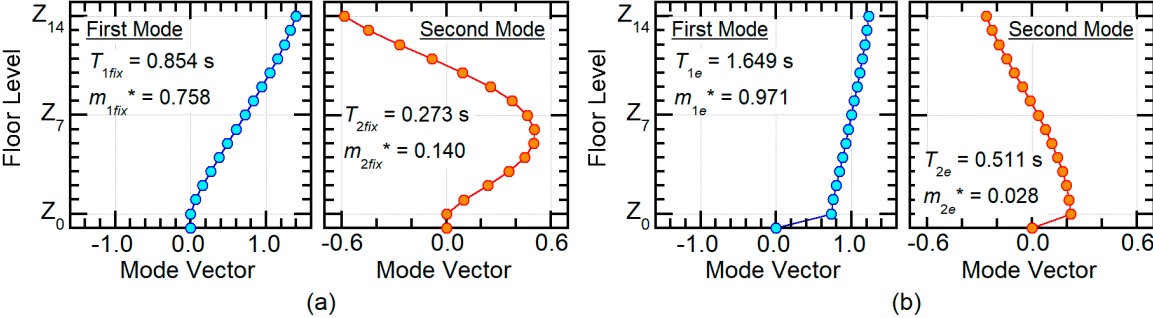

**Figure 6.** Shapes of the first and second natural mode vectors of the building models in the elastic range: (**a**) non-base-isolated; (**b**) base-isolated.

*3.2. Ground Motion Data*

The same two series of artificial ground motions used in the previous study by the authors [26] are used in this study to carry out the nonlinear time-history analysis. Figure 7 shows the pseudo acceleration spectrum of the generated artificial ground motions, and Figure 8 shows a representative example wave of the generated ground motions. The Art-S Series (waves Art-S00 to S11) are the artificial waves based on the horizontal major component of the 1995 JMA Kobe records, while Art-L series (waves Art-L00 to L11) are the artificial waves based on the horizontal major component of Sendai Government Office building #2 recorded during the 2011 earthquake, which affected the Pacific coast of Tohoku [30].

Because the cumulative strain energy of the LRBs and dampers are some of the target engineering parameters predicted in this study, the total input energy of the ground motion plays a crucial role in the study. Thus, the equivalent velocity of the total input energy for the SDOF model $V_I$, is calculated from the following equation for the measurement of the total input energy:

$$V_I = \sqrt{2 \int_0^{t_d} e_I(t) dt}, e_I(t) = -\dot{x}(t) a_g(t) \tag{18}$$

In Equation (18), $e_I(t)$ and $\dot{x}(t)$ are the energy rate and relative velocity of the SDOF model.

Figure 9 shows the elastic total input energy spectrum ($V_I$ spectrum) for the linear SDOF model with a damping ratio $h$ = 0.10, following the work by Akiyama [29]. As shown in Figure 9, $V_I$ of the Art-L series is approximately twice that of the Art-S series.

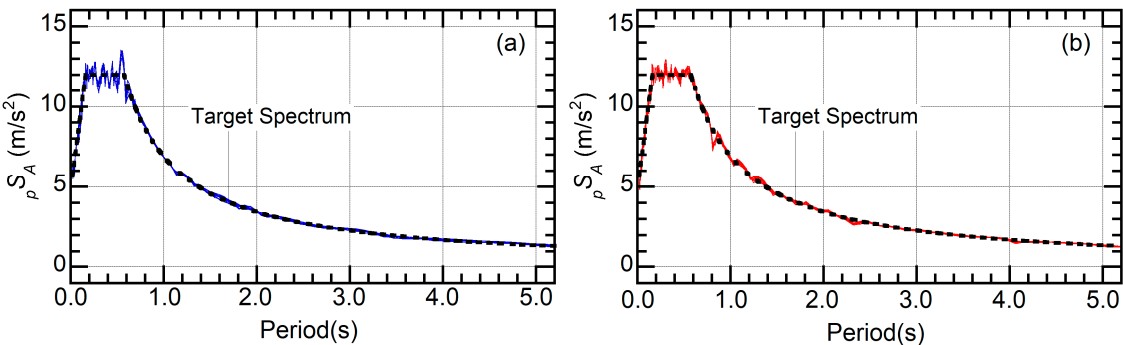

**Figure 7.** Elastic pseudo-acceleration target spectra of the generated artificial ground motions: (**a**) Art-S Series; (**b**) Art-L Series.

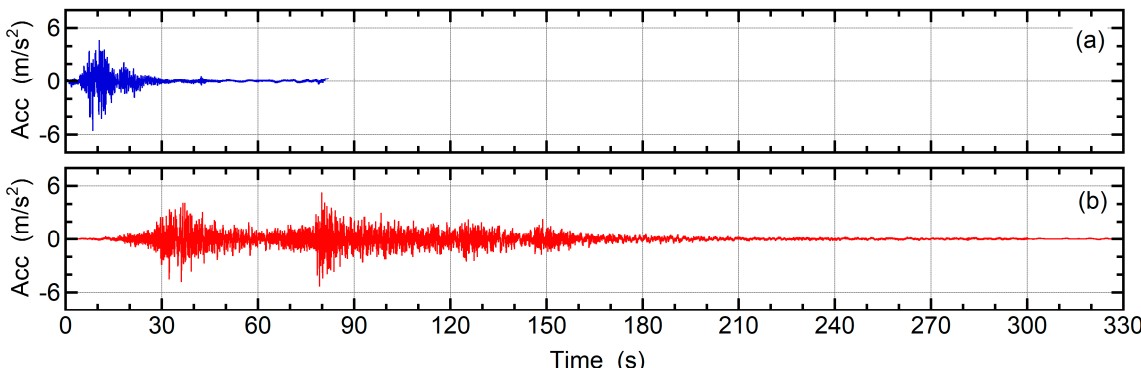

**Figure 8.** Example waves of the generated artificial ground motions: (**a**) Art-S00; (**b**) Art-L00.

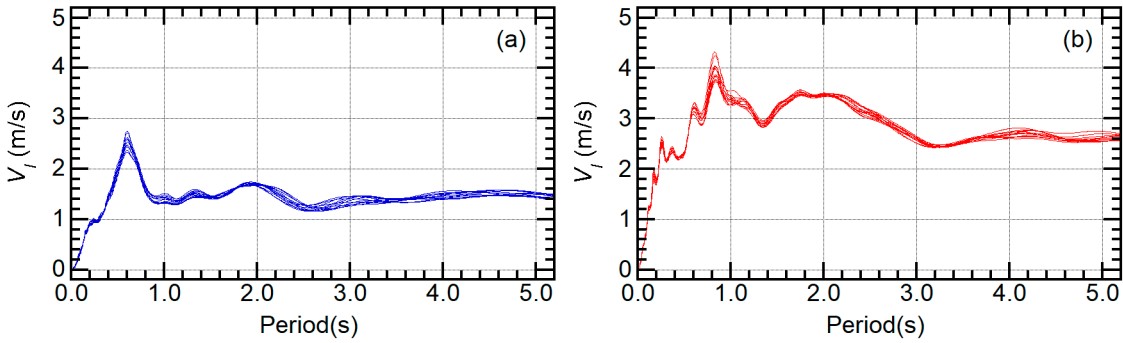

**Figure 9.** Elastic $V_I$ spectrum of the generated artificial ground motions: (**a**) Art-S Series; (**b**) Art-L Series.

### 3.3. Nonlinear Time-History Analysis Cases

In this study, the seismic excitation is considered as unidirectional in the X-direction. Nonlinear time-history analyses are carried out using the $2 \times 12 = 24$ artificial ground motions presented in Section 3.2. The intensities of ground motions are scaled to 50%, 75%, and 100%. Therefore, $3 \times 24 = 72$ analyses are carried out.

### 4. Analysis Results

In this section, the responses predicted by the procedure proposed in Section 2 are compared with the nonlinear time-history analysis results. The values $D_1{}^*_{max}$, $A_1{}^*_{max}$, $A_2{}^*_{max}$, and $V_{I1}{}^*$ used in the following prediction are considered as the mean of the results obtained from the nonlinear and linear time-history analyses for the equivalent SDOF models by considering 12 waves in each series.

### 4.1. Pushover Analysis Results

Figure 10 shows the $A_1{}^*$-$D_1{}^*$ relationship obtained from the pushover analysis result. In this study, the bi-linear idealization of the $A_1{}^*$-$D_1{}^*$ relationship is made according to the initial natural period $T_{1e}$ and the two points on the $A_1{}^*$-$D_1{}^*$ curve ($D_{1\ 0.10}^*$, $A_{1\ 0.10}^*$) and ($D_{1\ 0.40}^*$, $A_{1\ 0.40}^*$), respectively, where $A_{1\ 0.10}^*$ is the equivalent acceleration at $D_{1\ 0.10}^*$ (= 0.10 m) and $A_{1\ 0.40}^*$ is the equivalent acceleration at $D_{1\ 0.40}^*$ (= 0.40 m).

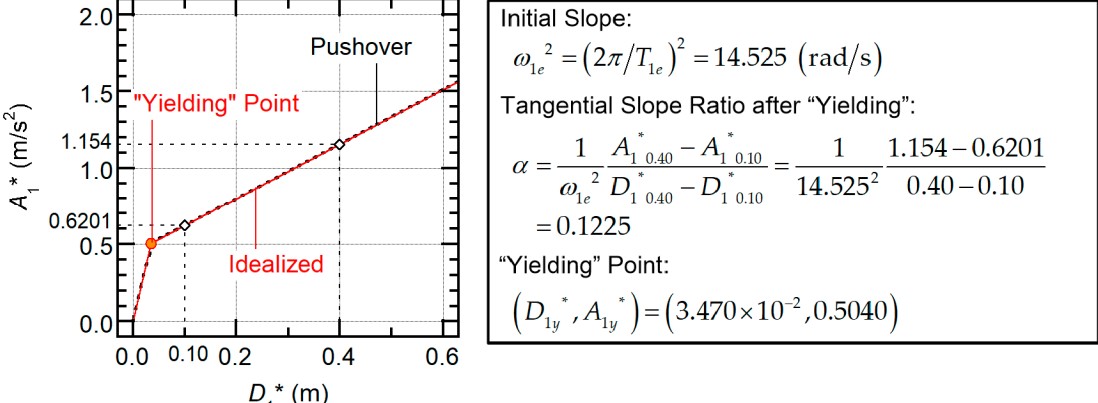

**Figure 10.** Bi-linear idealization of $A_1{}^*$-$D_1{}^*$ for the equivalent SDOF model representing the first modal response.

Figure 11 shows the influence of the change in the mode shape at each step. To evaluate the influence of the change in the mode shape at each step, the effective modal mass ratio of the two modes at each step are calculated as follows:

$$_n m_1^* = \frac{_n M_1^*}{M} = \frac{1}{M} \frac{\left(_n\mathbf{d}^T\mathbf{M1}\right)^2}{_n\mathbf{d}^T\mathbf{M}_n\mathbf{d}} , \, _n m_2^* = \frac{_n M_2^*}{M} = \frac{1}{M} \frac{\left(_n\boldsymbol{\varphi}_2{}^T\mathbf{M1}\right)^2}{_n\boldsymbol{\varphi}_2{}^T\mathbf{M}_n\boldsymbol{\varphi}_2} \tag{19}$$

$$_n\boldsymbol{\varphi}_2 = \boldsymbol{\varphi}_{2e} - \frac{\boldsymbol{\varphi}_{2e}{}^T\mathbf{M}_n\boldsymbol{\varphi}_1}{_n\boldsymbol{\varphi}_1{}^T\mathbf{M}_n\boldsymbol{\varphi}_1} {}_n\boldsymbol{\varphi}_1 = \boldsymbol{\varphi}_{2e} - \frac{\boldsymbol{\varphi}_{2e}{}^T\mathbf{M}_n\mathbf{d}}{_n\mathbf{d}^T\mathbf{M}_n\mathbf{d}} {}_n\mathbf{d} \tag{20}$$

As shown in Figure 11a, the deformation of the isolated layer becomes predominant as the equivalent displacement $D_1{}^*$ increases, and the components of all floors are approximately equal to unity when $D_1{}^* = 0.4$ m. This implies that, in the first mode response, the superstructure behaves as a rigid body. Figure 11b shows the variation of the effective modal mass for the first and second modes. As shown in this figure, the change of $m_1{}^*$ is relatively stable and closer to unity as $D_1{}^*$ increases. The change of $m_1{}^*$ occurs from 0.971 at first to 0.999 at $D_1{}^* = 0.4$ m. In contrast, the change of $m_2{}^*$ is significant; $m_2{}^*$ is 0.028 at first and drops to $9.340 \times 10^{-4}$ when $D_1{}^* = 0.4$ m. Figure 11c shows the change of the $m_2{}^*/m_{2e}{}^*$ ratio. As can be seen, the $m_2{}^*/m_{2e}{}^*$ ratio drastically drops as $D_1{}^*$ increases; the $m_2{}^*/m_{2e}{}^*$ ratio is 0.161 when $D_1{}^* = 0.1$ m, and 0.0329 when $D_1{}^* = 0.4$ m. Therefore, the change in the effective modal mass ratio of the second mode is significant for properly predicting the response of base-isolated buildings.

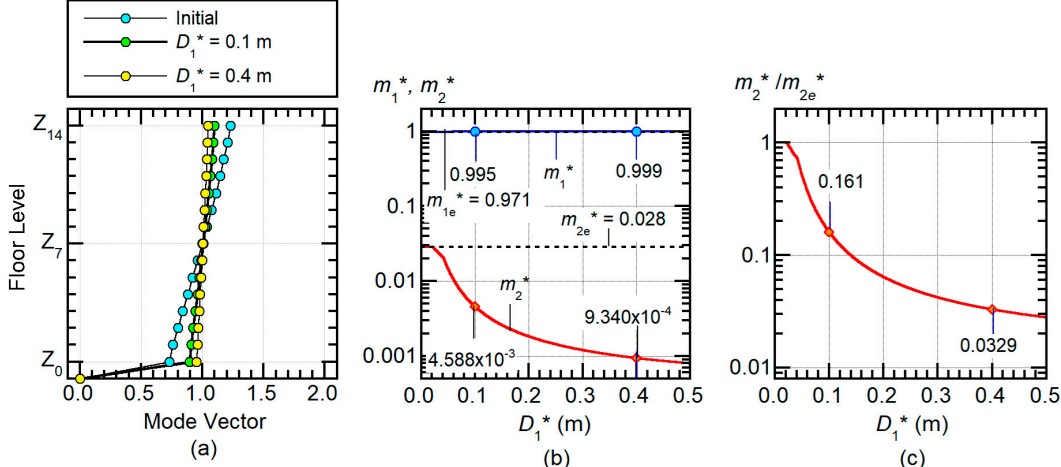

**Figure 11.** Influence of the mode shape change at each step: (**a**) variation of the first mode shape; (**b**) variation of the effective modal mass ratio for two modes; (**c**) variation of $m_2^*/m_{2e}^*$.

*4.2. Validation of the Predicted Results*

4.2.1. Maximum Floor Response

Figures 12 and 13 present the comparisons of the maximum floor response (relative displacement and absolute acceleration) relevant to the Art-S series and Art-L series, respectively. The average, maximum, and minimum results of the nonlinear time-history analysis are shown for each series and ground motion intensity. In these figures, the predicted results considering only the first mode response and those considering the second mode response are denoted as "First Mode" and "Envelope," respectively.

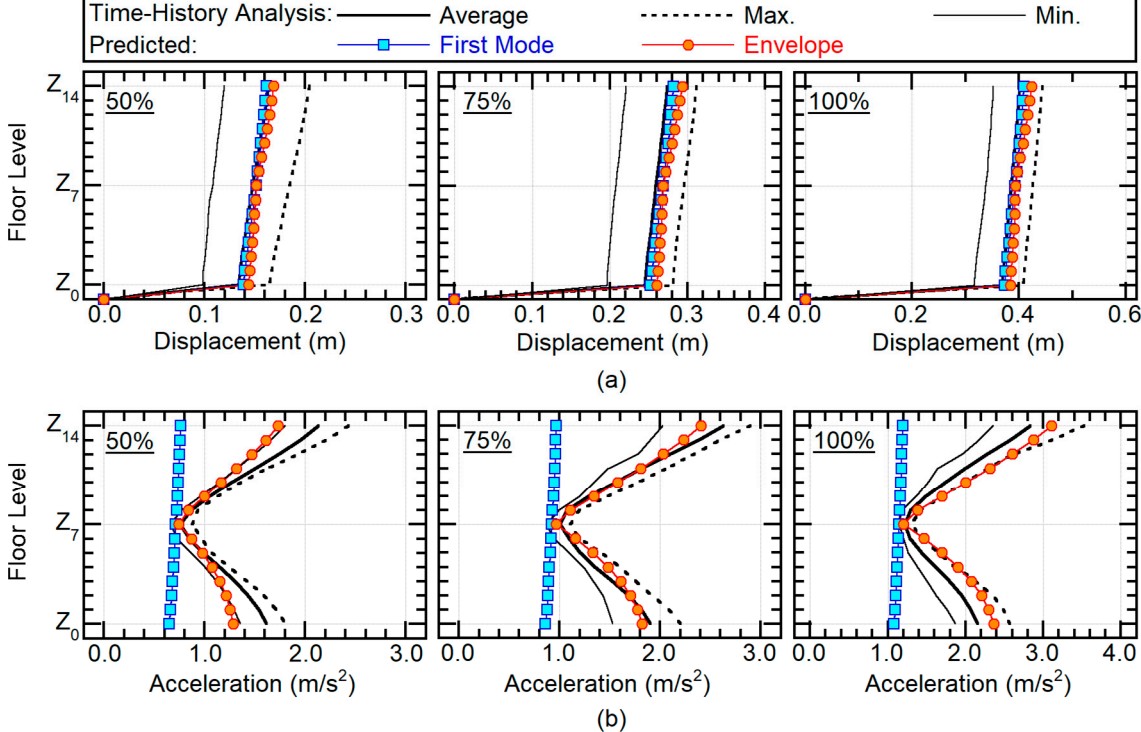

**Figure 12.** Comparison of maximum floor response (Art-S series): (**a**) relative displacement; (**b**) absolute acceleration.

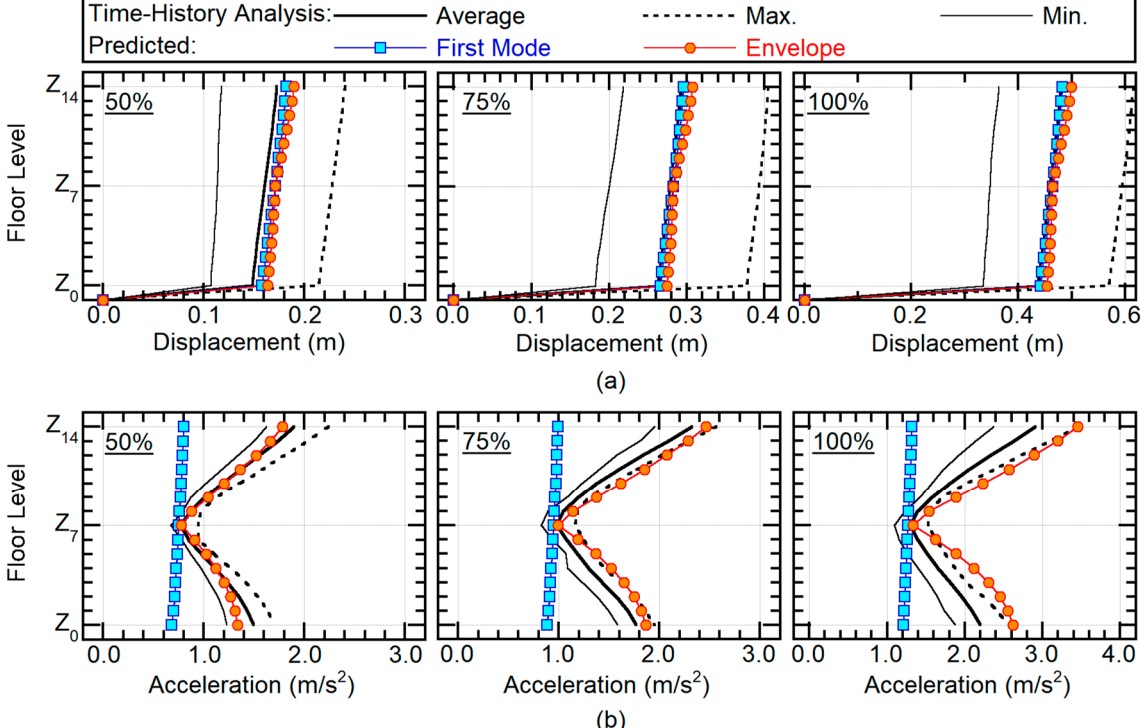

**Figure 13.** Comparisons of maximum floor response (Art-L series): (**a**) relative displacement; (**b**) absolute acceleration.

As shown in Figures 12a and 13a, the predicted relative displacements obtained by considering only the first mode are in good agreement with the average time-history analysis results. The predicted results shown as the envelope are also accurate, and the difference between the two predicted results is small.

However, as shown by diagrams in Figures 12b and 13b, the absolute acceleration predicted by considering only the first mode significantly underestimates the average time-history analysis results. In contrast, the predicted results presented as "envelope" (obtained taking into account the second mode contribution) are much closer to the time-history analysis results. The difference between the two predicted results is significant at the $Z_0$ and $Z_{14}$ levels, that is, the lowest and highest floors. These figures also show that the maximum floor acceleration predicted by the envelope slightly underestimates the time-history analysis results when the ground motion intensity is 50% and overestimates it when the ground motion intensity is 100%.

### 4.2.2. Maximum Shear Forces of Vertical Members

Figures 14 and 15 present the comparisons between the maximum shear forces of the vertical members. The results at (a) column $X_2Y_1$ and (b) column $X_2Y_2$ (with the spandrel wall) are provided in the figures. As can be seen, the results predicted by considering only the first mode significantly underestimate the average time-history analysis results. However, the predicted results presented as envelopes are in good agreement with the average time-history analysis results when the ground motion intensity is 50% or 75%, and conservative when the ground motion intensity is 100%.

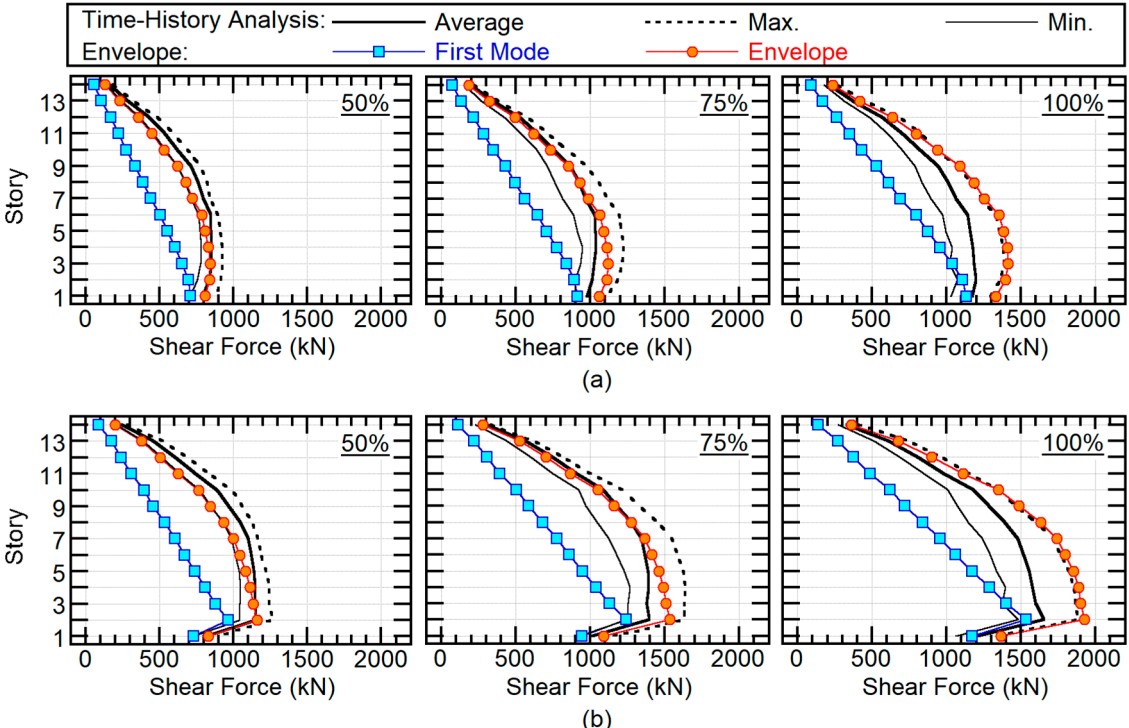

**Figure 14.** Comparisons between the maximum shear forces of vertical members (Art-S series): (**a**) column $X_2Y_1$ and (**b**) column $X_2Y_2$ (with spandrel walls).

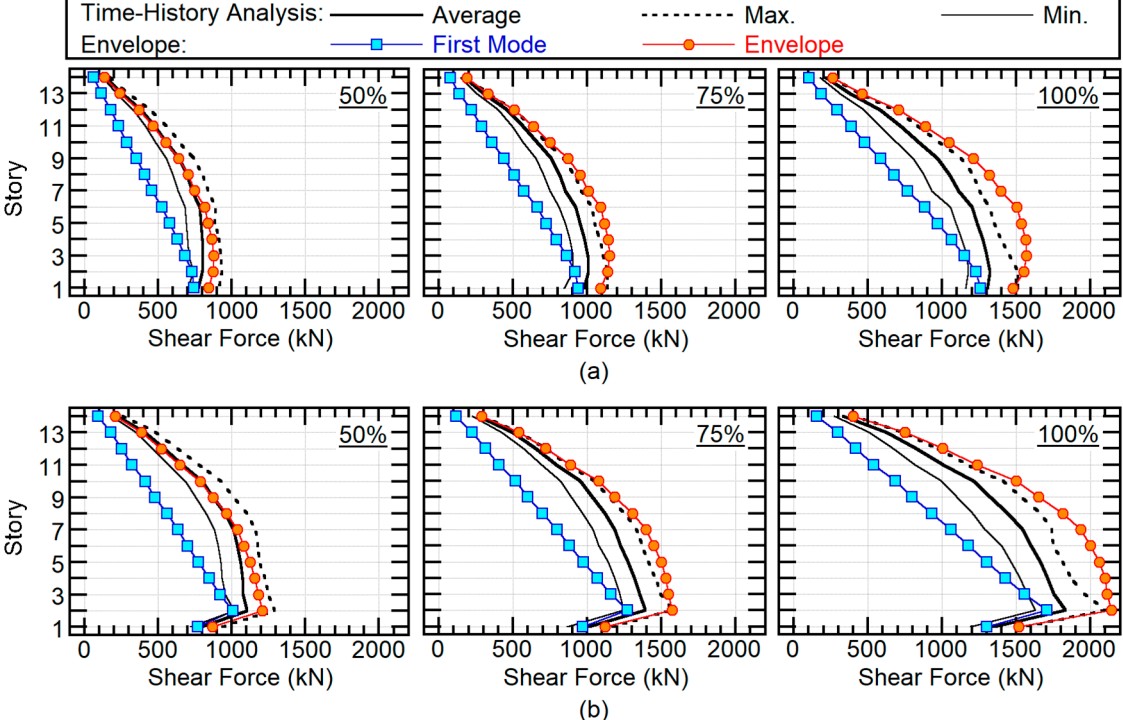

**Figure 15.** Comparisons between maximum shear forces of vertical members (Art-L series): (**a**) column $X_2Y_1$ and (**b**) column $X_2Y_2$ (with spandrel walls).

### 4.2.3. Maximum Response of Isolators

Figure 16 shows the comparisons between the maximum responses of the isolators (NRB at $X_1Y_1$, LRB at $X_1Y_2$). The relationship between the maximum shear strain $\gamma$ and the maximum and minimum

compression stress σ is shown. In this figure, the "ultimate compressive stress" is the final ultimate property obtained from a catalog provided by the Bridgestone Corporation [31]. Notably, the peak shear strain $\gamma$ and the compressive stress σ obtained from the time-history analysis results are the same as those obtained from the average nonlinear time-history analysis results for 12 waves in each series, without consideration to the simultaneity of the peak responses.

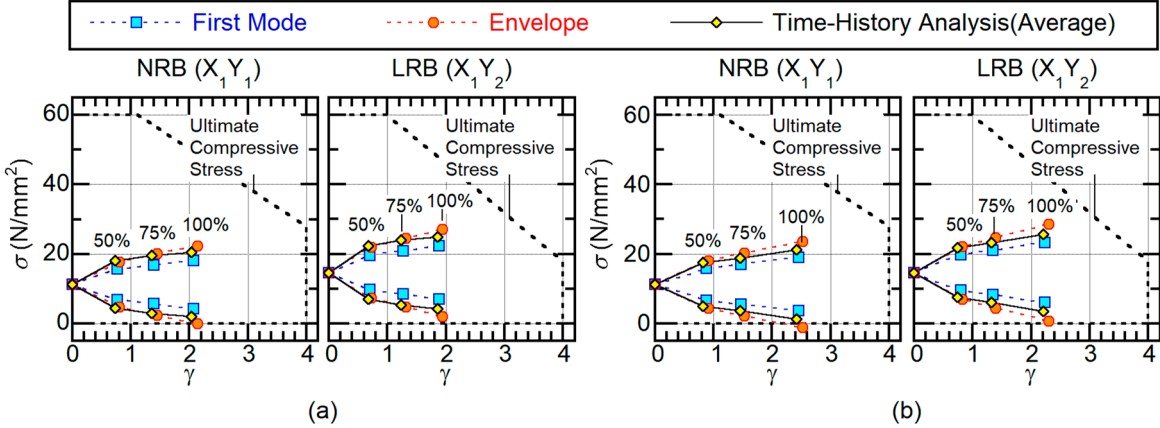

**Figure 16.** Comparisons between maximum responses of isolators: (**a**) Art-S series; (**b**) Art-L series.

As shown in Figure 16, the results predicted by the envelope are slightly conservative compared to the time-history analysis results, while the results predicted only by the first mode are not conservative. The figure also shows that the plots are within the ultimate properties of those isolators, except for the response of isolator $X_1Y_1$ obtained from the envelope when the ground motion intensity is 100%. However, the nominal stress of isolator $X_1Y_1$ (and isolator $X_1Y_2$) obtained from all time-history analysis results is within the compression range. Therefore, the behavior of all isolators satisfies the given ultimate properties under the given ground motion intensities.

4.2.4. Cumulative Strain Energy of Isolators and Dampers

Figure 17 presents the comparisons between the cumulative strain energies of the isolators (LRBs) and steel dampers. As prescribed in Section 2, the predicted cumulative strain energies are based only on the first modal response.

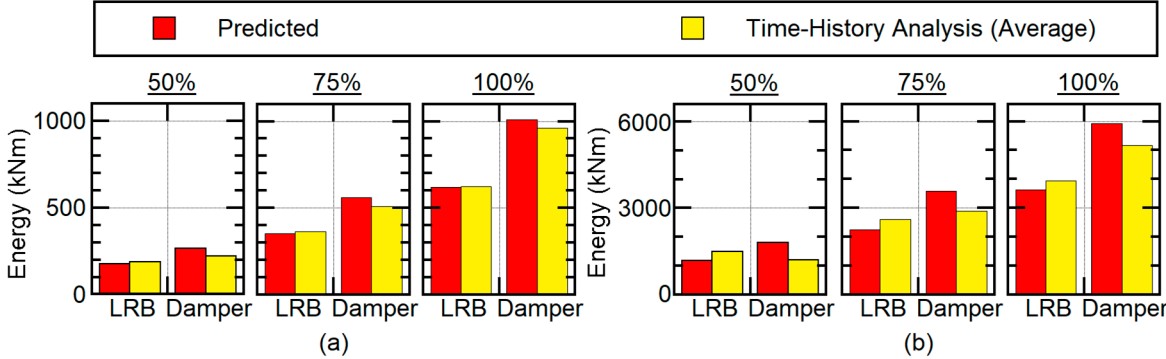

**Figure 17.** Comparisons of cumulative strain energy between isolators and dampers: (**a**) Art-S series; (**b**) Art-L series.

As shown in Figure 17a, the predicted cumulative strain energies are in good agreement with that obtained from the average time-history analysis results in the case of the Art-S series. The predicted cumulative energy of the LRBs is slightly smaller than the time-history results, while the predicted cumulative energy of the steel damper is slightly larger compared with the time-history results. In the

case of the Art-L series, the predicted results are similar to the time-history results; however, their difference is larger compared with the results in the case of the Art-S series.

## 5. Discussion

This section discusses the accuracy in predicting (i) the maximum modal response, and (ii) the cumulative response.

### 5.1. Accuracy in Predicting Maximum Modal Responses

Figure 18 shows the comparisons between the maximum modal responses obtained from the nonlinear time-history analysis of the frame building model (MDOF) and obtained by the equivalent SDOF models. Here, the modal responses of the MDOF model are calculated using the procedure proposed in a previous paper by the authors [26].

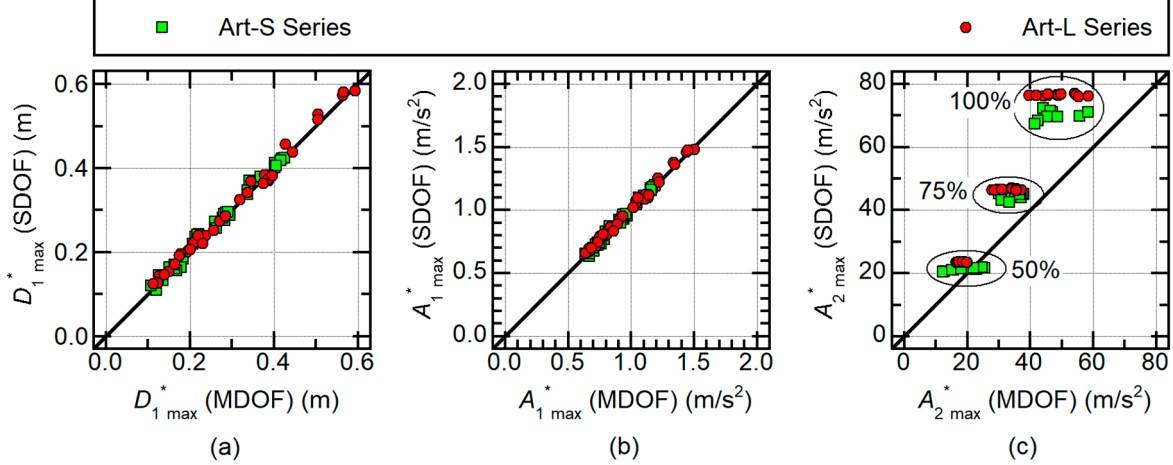

**Figure 18.** Comparison of maximum modal responses: (**a**) equivalent displacement of first mode; (**b**) equivalent acceleration of first mode; (**c**) equivalent acceleration of second mode.

As shown in Figure 18a,b, the predicted maximum response of the first mode ($D_1{}^*{}_{max}$ and $A_1{}^*{}_{max}$) by the equivalent SDOF model is in very good agreement with that of the MDOF model. However, according to the second mode, the predicted maximum acceleration of the second modal response by the equivalent SDOF model is conservative. The results obtained by the equivalent SDOF model are in good agreement with those obtained by the MDOF model in the case wherein the ground motion intensity is 50%. However, the difference between the results obtained by the equivalent SDOF and MDOF models is larger in the cases wherein the ground motion intensities are 75% and 100%.

Figure 19 shows the comparisons between the first modal responses calculated from the time-history analysis of the MDOF and SDOF models (Art-L00, 100%). As shown in this figure, the hysteresis behavior ($A_1{}^*$-$D_1{}^*$ relationship) and the time histories obtained by the equivalent SDOF model are in very good agreement with those obtained by the MDOF model.

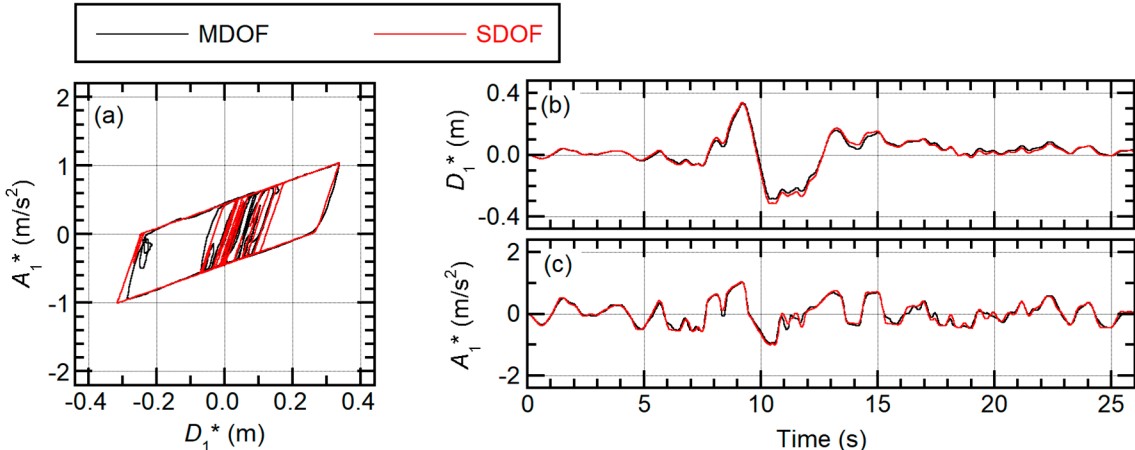

**Figure 19.** Comparisons between first modal response calculated from time-history analysis of multi-degree-of freedom (MDOF) and SDOF models (Art-L00, 100%): (**a**) $A_1^*$-$D_1^*$ relationship comparisons; (**b**) time-history of $D_1^*$; (**c**) time-history of $A_1^*$.

Figure 20 shows the comparisons between the time histories of the second modal acceleration $A_2^*(t)$ for two different ground motion intensities (Art-S06). As shown in Figure 20a, the time-history obtained by the equivalent SDOF model is in very good agreement with that by the MDOF model when the ground motion intensity is 50%. However, the difference of the time-history becomes significant when the ground motion intensity is 100%, as shown in Figure 20b.

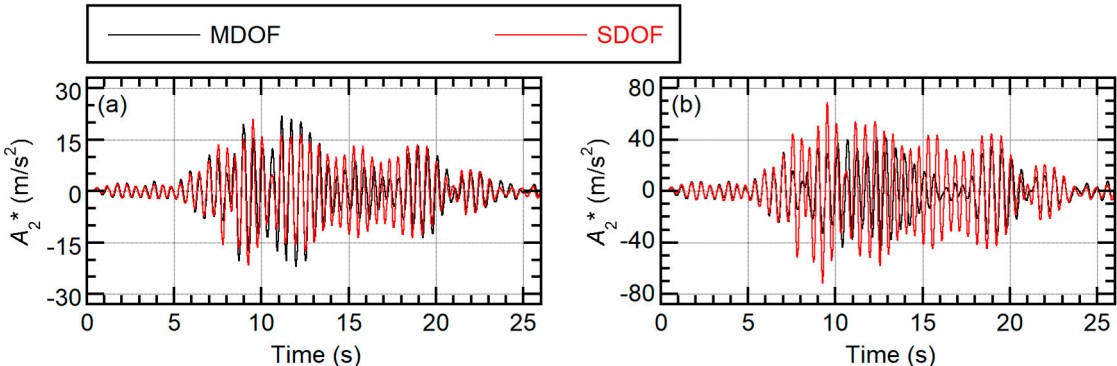

**Figure 20.** Time-history comparison of second modal acceleration: (**a**) Art-S06, 50%; (**b**) Art-S06, 100%.

To investigate the time-history difference of $A_2^*(t)$ obtained from the equivalent SDOF and MDOF models, Fourier transform analysis is carried out for $A_2^*(t)$. Figure 21 presents the comparisons between the Fourier amplitudes of the second modal acceleration for two different ground motion intensities. In this figure, the natural frequency of the second modal response ($f_{2e}$ = 1.957 Hz) is also shown. Moreover, as shown in Figure 21a, the Fourier amplitude distributions in the results obtained by each of the two models are in good agreement when the ground motion intensity is 50%. Additionally, when the ground motion intensity is 100%, as shown in Figure 21b, the results are similar, although the predominant frequency is approximately 1.8 Hz. Therefore, the validity of the assumption of the second modal response being approximated as a linearly elastic response with consideration to the change in the effective second modal mass is confirmed. The conservative evaluation of $A_2^*{}_{\max}$ may have arisen from the energy absorption of the second modal response, owing to the hysteresis response of the isolation layer. In the linear analysis of the equivalent SDOF model, only the elastic viscous damping is considered.

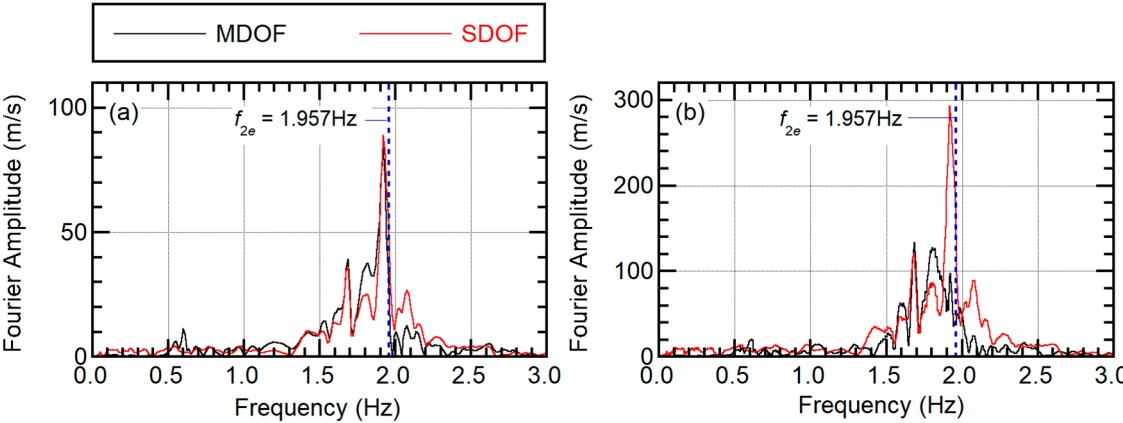

**Figure 21.** Comparisons of Fourier amplitudes of second modal acceleration: (**a**) Art-S06, 50%; (**b**) Art-S06, 100%.

*5.2. Accuracy in Predicting Cumulative Responses*

Figure 22 presents the comparisons between the equivalent velocities of the cumulative energy input of the first mode, $V_{I1}{}^*$, obtained by the equivalent SDOF and MDOF models. In this figure, both results were obtained from the average of 12 waves in each series. The results of the equivalent SDOF model agree with the results of the MDOF model. Therefore, the accuracy of the predicted $V_{I1}{}^*$ is satisfactory.

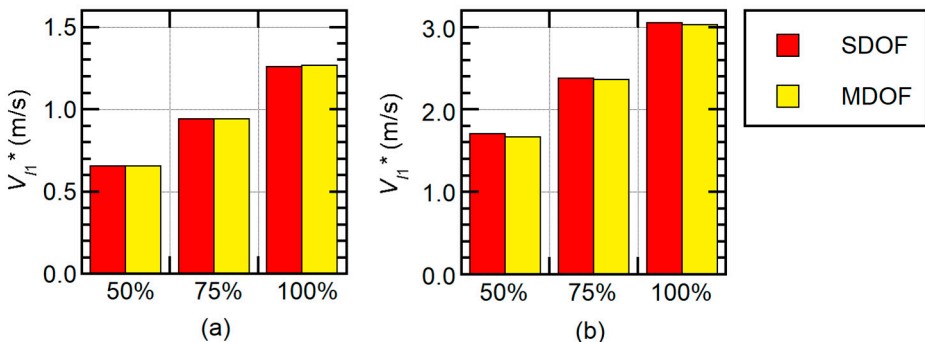

**Figure 22.** Comparisons of equivalent velocities of cumulative energy input for first mode: (**a**) Art-S series; (**b**) Art-L series.

To evaluate the accuracy difference between the predicted cumulative energies of the LRBs and the dampers, the equivalent number of cycles $n_d$ is defined as follows:

$$n_{dk} = \frac{E_{SDk}}{4Q_{yDk}\left(\delta_{Dk\max} - \delta_{yDk}\right)},\tag{21}$$

where $\delta_{Dk\max}$ (= $x_{0\max}$) is the maximum deformation of the $k$th dampers.

Figure 23 presents the comparisons between the $n_d$ values of the LRBs and the dampers. The time-history analysis results presented in this figure were calculated according to the average $E_{SDk}$ and $\delta_{D\max}$. As shown in Figure 23a, the difference between the predicted $n_d$ and the time-history analysis results is small. In contrast, the difference between the two results is significant in the case of the Art-L series, as shown in Figure 23b. The $n_d$ values predicted for the LRBs and dampers have the same value, while those obtained from the time-history analysis are different. For the LRBs, the $n_d$ prediction is underestimated, while $n_d$ prediction for the dampers is overestimated. The $n_d$ difference between the LRBs and the dampers observed in the time-history analysis results is attributed to the

difference of their yield deformation. Notably, the yield deformation of the LRB was smaller ($\delta_{yD} = 1.95 \times 10^{-2}$ m) than that of the steel damper ($\delta_{yD} = 3.17 \times 10^{-2}$ m).

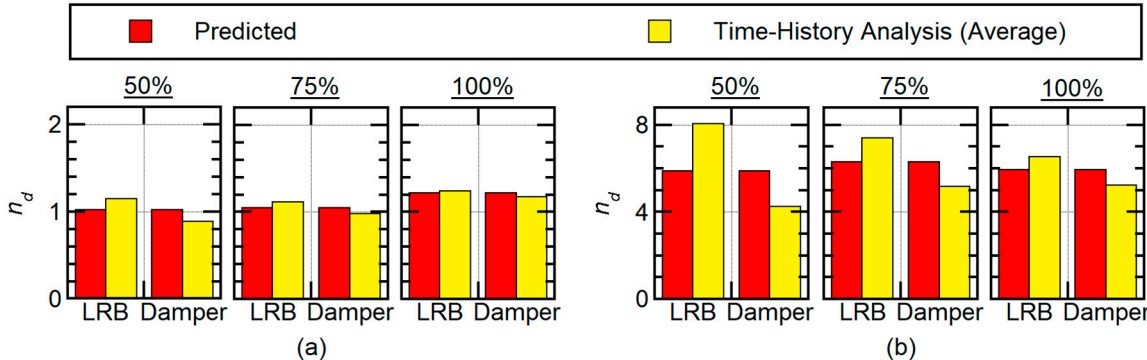

**Figure 23.** Comparisons between equivalent numbers of cycles for LRBs and dampers: (**a**) Art-S series; (**b**) Art-L series.

## 6. Conclusions

This study predicted the maximum and cumulative seismic response of a 14-story reinforced concrete base-isolated frame building through the pushover analysis and nonlinear time-history analysis of the equivalent SDOF models. The maximum equivalent acceleration of the second mode, which is used in the pushover analysis, is predicted considering the change of the second effective modal mass due to the nonlinearity of the isolation layer by using the equation proposed in this study. The engineering parameters predicted in this study are as follows: (i) maximum relative floor displacement, (ii) maximum absolute floor acceleration, (iii) maximum shear forces of vertical member in the superstructure, (iv) maximum shear strain and nominal stress of isolator, and (v) cumulative strain energy of LRBs and dampers. The main contributions of and conclusions drawn from this study are as follows:

- The consideration of the second modal response's contribution is important to better predict the maximum floor acceleration and maximum shear forces of the vertical members in the superstructure. The predicted maximum response obtained by the proposed procedure is in good agreement with the nonlinear time-history analysis results.
- The cumulative strain energy of the isolators (LRBs) and dampers in the isolation layer can be satisfactorily predicted by considering only the first modal response. The reason for this is that the effective first modal mass is approximately 100% of the total mass.
- The maximum and cumulative response of the first mode is satisfactorily predicted using the equivalent SDOF model.
- The maximum equivalent acceleration of the second modal response is conservatively predicted by magnifying the linear analysis results of the equivalent SDOF model and considering the change in the equivalent second modal mass.

Therefore, according to the base-isolated frame building investigated herein, the authors believe that all engineering parameters considered in this study can be successfully predicted using the proposed procedure. However, this conclusion strongly relies on the assumption that the behavior of the superstructure is perfectly linear elastic. Even for a newly designed building, stiffness degradation may occur owing to the cracking of reinforced concrete members. In such cases, the proposed procedure can be applied by considering the pre-determined reduction of the member stiffness in the superstructure. Thus, further studies are needed to investigate the applicability of the proposed procedure. The authors think that the ratio of the isolated period $T_f$ to $T_{1fix}$ (the fundamental period of the superstructure, assuming it is non-base-isolated) is an important factor to investigate. In the case wherein $T_f/T_{1fix}$ is small, or $T_f$ is not sufficiently separated from $T_{1fix}$, the proposed procedure may be

difficult to apply. The evaluation of both the maximum and cumulative energy of each modal response without carrying out time-history analysis for the equivalent SDOF model is another important issue. For design purposes, it is very useful that both responses are evaluated based on the response spectrums. Notably, the maximum response can be easily estimated from the predetermined design response spectrum. However, evaluating the cumulative response is more difficult because, thus far, a design cumulative energy spectrum has not been implemented in the design code. Therefore, the authors think that a procedure should be developed to estimate both the maximum and cumulative responses from the ground motion characteristics, which highlights another important issue in this study.

**Author Contributions:** Conceptualization and methodology, K.F., Y.M., and T.N.; software and validation, K.F.; modelling of buildings and analysis, data curation, Y.M. and T.N.; analysis supervision and data checking, K.F.; writing-original draft preparation in Japanese, Y.M. and T.N.; writing-English writing, K.F. All authors have read and agreed to the final version of the manuscript.

**Funding:** This study did not receive external funding.

**Acknowledgments:** We thank Edanz Group (https://en-author-services.edanzgroup.com/) for editing a draft of this manuscript.

**Conflicts of Interest:** The authors declare that there is no conflict of interest.

## Appendix A. Magnification Factor for Equivalent Second Modal Acceleration

Let Equation (A1) be the motion equation for an *N*-story base-isolated building model.

$$\mathbf{M}\ddot{\mathbf{d}}(t) + \mathbf{C}\dot{\mathbf{d}}(t) + \mathbf{f_R}(t) = -\mathbf{M}1a_g(t) \tag{A1}$$

It is assumed that the restoring force vector $\mathbf{f_R}(t)$ can be expressed as follows:

$$\mathbf{f_R}(t) = \mathbf{f_{R1}}(t) + \mathbf{f_{R2h}}(t) \tag{A2}$$

$$\mathbf{f_{R1}}(t) = \mathbf{M}(\Gamma_{1ie}\boldsymbol{\varphi_{1ie}})A_1{}^*(t) \tag{A3}$$

In Equation (A3), $\mathbf{f_{R1}}(t)$ is the restoring force vector of the first modal response, and vector $\Gamma_{1ie}\boldsymbol{\varphi_{1ie}}$ is the first mode vector corresponding to the maximum equivalent displacement of the first mode, $D_1{}^*_{\max}$. It is assumed that the restoring force vector of the second and higher modal response, $\mathbf{f_{R2h}}(t)$, can be approximated as follows:

$$\mathbf{f_{R2h}}(t) = \mathbf{f_R}(t) - \mathbf{f_{R1}}(t) \approx \mathbf{M}(\Gamma_{2ie}\boldsymbol{\varphi_{2ie}})A_2{}^*(t) \tag{A4}$$

where vector $\Gamma_{2ie}\boldsymbol{\varphi_{2ie}}$ is the second mode vector corresponding to $D_1{}^*_{\max}$. Additionally, it is assumed that the equivalent acceleration of the second mode, in terms of the second mode vector in the elastic range, $A_{2e}{}^*(t)$, can be approximated as follows:

$$A_{2e}{}^*(t) = \frac{\Gamma_{2e}\boldsymbol{\varphi_{2e}}^{\mathbf{T}}\mathbf{f_R}(t)}{M_{2e}{}^*} \approx \frac{\Gamma_{2e}\boldsymbol{\varphi_{2e}}^{\mathbf{T}}\mathbf{f_{R2h}}(t)}{M_{2e}{}^*} \tag{A5}$$

$$M_{2e}{}^* = \Gamma_{2e}{}^2\boldsymbol{\varphi_{2e}}^{\mathbf{T}}\mathbf{M}\boldsymbol{\varphi_{2e}} = \frac{\left(\boldsymbol{\varphi_{2e}}^{\mathbf{T}}\mathbf{M}1\right)^2}{\boldsymbol{\varphi_{2e}}^{\mathbf{T}}\mathbf{M}\boldsymbol{\varphi_{2e}}} \tag{A6}$$

In Equations (A5) and (A6), vector $\Gamma_{2e}\boldsymbol{\varphi_{2e}}$ is the second mode vector in the elastic range, and $M_{2e}{}^*$ is the effective modal mass of the second mode in the elastic range.

The following discussion focuses on the relationship between the two equivalent modal accelerations, $A_{2e}{}^*(t)$ and $A_2{}^*(t)$, and the $A_2{}^*(t)/A_{2e}{}^*(t)$ ratio is the magnification factor resulting from the change in the second mode shape.

First, the mode vectors $\boldsymbol{\varphi}_{\mathbf{1ie}}$, $\boldsymbol{\varphi}_{\mathbf{2ie}}$, and $\boldsymbol{\varphi}_{\mathbf{2e}}$ are normalized as follows:

$$\boldsymbol{\varphi}_{\mathbf{1ie}}{}^{\mathbf{T}}\mathbf{M}\boldsymbol{\varphi}_{\mathbf{1ie}} = \boldsymbol{\varphi}_{\mathbf{2ie}}{}^{\mathbf{T}}\mathbf{M}\boldsymbol{\varphi}_{\mathbf{2ie}} = \boldsymbol{\varphi}_{\mathbf{2e}}{}^{\mathbf{T}}\mathbf{M}\boldsymbol{\varphi}_{\mathbf{2e}} = \mathbf{1}^{\mathbf{T}}\mathbf{M}\mathbf{1} = M \tag{A7}$$

where $M$ is the total mass of the superstructure above the isolated layer. Therefore, the modal participation factors and the effective modal mass are expressed as follows:

$$\Gamma_{1ie} = \frac{\boldsymbol{\varphi}_{\mathbf{1ie}}{}^{\mathbf{T}}\mathbf{M}\mathbf{1}}{M}, \Gamma_{2ie} = \frac{\boldsymbol{\varphi}_{\mathbf{2ie}}{}^{\mathbf{T}}\mathbf{M}\mathbf{1}}{M}, \Gamma_{2e} = \frac{\boldsymbol{\varphi}_{\mathbf{2e}}{}^{\mathbf{T}}\mathbf{M}\mathbf{1}}{M} \tag{A8}$$

$$M_{1ie}{}^{*} = \Gamma_{1ie}{}^{2}M, M_{2ie}{}^{*} = \Gamma_{2ie}{}^{2}M, M_{2e}{}^{*} = \Gamma_{2e}{}^{2}M \tag{A9}$$

Next, the second mode vector $\boldsymbol{\varphi}_{\mathbf{2ie}}$ is assumed to be expressed as follows:

$$\boldsymbol{\varphi}_{\mathbf{2ie}} = c_1\boldsymbol{\varphi}_{\mathbf{1ie}} + c_2\boldsymbol{\varphi}_{\mathbf{2e}}. \tag{A10}$$

Based on the abovementioned preparations, the relationship between $A_2{}^{*}(t)$ and $A_{2e}{}^{*}(t)$ can be derived. From the orthogonal condition of mode vectors $\boldsymbol{\varphi}_{\mathbf{1ie}}$ and $\boldsymbol{\varphi}_{\mathbf{2ie}}$, the ratio of constant $c_1$ to constant $c_2$ can be expressed as follows:

$$c_{12} \equiv \frac{c_1}{c_2} = -\frac{\boldsymbol{\varphi}_{\mathbf{1ie}}{}^{\mathbf{T}}\mathbf{M}\boldsymbol{\varphi}_{\mathbf{2e}}}{M} \tag{A11}$$

By substituting Equation (A11) into Equation (A10), the following relationship holds:

$$\begin{aligned}\boldsymbol{\varphi}_{\mathbf{2ie}}{}^{\mathbf{T}}\mathbf{M}\boldsymbol{\varphi}_{\mathbf{2ie}} &= (c_1\boldsymbol{\varphi}_{\mathbf{1ie}} + c_2\boldsymbol{\varphi}_{\mathbf{2e}})^{\mathbf{T}}\mathbf{M}(c_1\boldsymbol{\varphi}_{\mathbf{1ie}} + c_2\boldsymbol{\varphi}_{\mathbf{2e}}) \\ &= (c_1{}^2 + c_2{}^2)M + 2c_1c_2\boldsymbol{\varphi}_{\mathbf{1ie}}{}^{\mathbf{T}}\mathbf{M}\boldsymbol{\varphi}_{\mathbf{2e}} \\ &= M\end{aligned} \tag{A12}$$

Considering Equation (A11), Equation (A12) can be rewritten as follows:

$$c_2{}^2(1 - c_{12}{}^2) = 1 \tag{A13}$$

Therefore, the constant $c_2$ can be expressed as follows:

$$c_2 = 1/\sqrt{1 - c_{12}{}^2} \tag{A14}$$

By substituting Equation (A14) into (A10), vector $\boldsymbol{\varphi}_{\mathbf{2e}}$ can be expressed as follows:

$$\boldsymbol{\varphi}_{\mathbf{2e}} = \sqrt{1 - c_{12}{}^2}\boldsymbol{\varphi}_{\mathbf{2ie}} + c_{12}\boldsymbol{\varphi}_{\mathbf{1ie}} \tag{A15}$$

By substituting Equations (A4) and (A15) into (A5), $A_{2e}{}^{*}(t)$ can be expressed as follows:

$$A_{2e}{}^{*}(t) \approx \frac{\Gamma_{2e}\boldsymbol{\varphi}_{\mathbf{2e}}{}^{\mathbf{T}}\mathbf{f}_{\mathbf{R2h}}(t)}{M_{2e}{}^{*}} = \frac{\Gamma_{2e}\Gamma_{2ie}\left(\sqrt{1 - c_{12}{}^2}\boldsymbol{\varphi}_{\mathbf{2ie}} + c_{12}\boldsymbol{\varphi}_{\mathbf{1ie}}\right)^{\mathbf{T}}\mathbf{M}\boldsymbol{\varphi}_{\mathbf{2ie}}}{M_{2e}{}^{*}}A_2{}^{*}(t) \tag{A16}$$

Considering the orthogonal condition of mode vectors $\boldsymbol{\varphi}_{\mathbf{1ie}}$ and $\boldsymbol{\varphi}_{\mathbf{2ie}}$, and Equation (A9), Equation (A16) can be simplified as follows:

$$A_{2e}{}^{*}(t) \approx \sqrt{1 - c_{12}{}^2}\sqrt{\frac{M_{2ie}{}^{*}}{M_{2e}{}^{*}}}A_2{}^{*}(t) \tag{A17}$$

Assuming that the square of constant $c_{12}$ (Equation (A11)) is negligibly small, a simpler relationship between $A_{2e}{}^*(t)$ and $A_2{}^*(t)$ can be obtained as follows:

$$A_2{}^*(t) \approx \sqrt{\frac{M_{2e}{}^*}{M_{2ie}{}^*}} A_{2e}{}^*(t) \tag{A18}$$

Equation (A18) implies that the equivalent acceleration of the second mode is magnified, owing to the variation of the effective modal mass in the nonlinear stage. In the proposed procedure, the equivalent acceleration $A_{2e}{}^*$ on the left side of Equation (A18) is calculated from the linear analysis of the equivalent SDOF model by assuming the natural period as $T_{2e}$ and viscous damping ratio as $h_{2e}$. Therefore, the change of the natural period and energy absorbing effect caused by the nonlinearity of the base isolation layer are not considered in the calculation of $A_{2e}{}^*$.

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
