# Peer review of "Predicting Maximum and Cumulative Response of A Base-isolated Building Using Pushover Analysis"

_buildings, doi:10.3390/buildings10050091_

Round 1

Reviewer 1 Report

The paper presents a numerical study to apply an analytical method based on pushover equivalent analyses. The method seems to have been already proposed in a previous paper of the authors. Validation of the method is given, by comparing the results of equivalent pushover analyses with those of non-linear time-history analyses carried out with two sets of artificial earthquakes. The maximum and cumulative responses are obtained, while different parameters are considered to assess the effectiveness of the method. The results show a good performance of the proposed method, although some of the assumptions made can be not realistic enough and the procedure may be not easy to apply in practice.

The following remarks should be drawn:

C1) The paper is written in poor English. The help of a native speaker is recommended. Some suggestions, which are not exhaustive though, are given in the attached file.

C2) The authors should highlight better (in the Introduction and in the Conclusions sections) what is the original contribution of the present paper with respect to a previous paper published by one of the authors in Buildings journal (2019). Are they presenting an application to a multi-story building in order to validate the method proposed in their previous paper? Of course, also a case-study that proves the validity of the method can be considered a valuable scientific contribution provided that it is clearly specified.

C3) The Introduction and the Bibliography should be improved by referring to other kinds of isolation systems, like partial mass isolation methods and isolation systems equipped with tuned-mass-dampers. Reference to papers [RIF1-RIF4] is recommended.

C4) References [REF5-REF7] can be cited when evidencing the power of non-linear time-history analyses to assess the seismic behavior of different kinds of structures (rows 49-50).

C5) Other comments and suggestions are given in the attached file.

References suggested:

[RIF1]  Porcu M. C. (2019) Partial floor mass isolation to control seismic stress in framed buildings. International Journal of Safety and Security Engineering9(2), 157-165.

[RIF2]  Anajafi H, Medina RA (2018) Comparison of the seismic performance of a partial mass isolation technique with conventional TMD and base-isolation systems under broad-band and narrow-band excitations. Engineering Structures, 158, 110-123.

[RIF3]  Hashimoto, T., Fujita, K., Tsuji, M. and Takewaki I. (2015) Innovative base-isolated building with large mass-ratio TMD at basement. Int. J. Future Cities Environ. 1, 9, 2015.

[RIF4]  De Domenico D, Ricciardi G. (2018) An enhanced base isolation system equipped with optimal tuned mass damper inerter (TMDI). J. Earth. Eng. & Struct. Dynamics, 47:1169-1192.

[REF5] M.C. Porcu, C.Bosu & I. Gavrić. (2018) Non-linear dynamic analysis to assess the seismic performance of cross-laminated timber structures. Journal of Building Engineering, 19, 480-493.

[REF6] Estekanchi, H. E., Riahi, H. T., & Vafai, A. (2011). Application of endurance time method in seismic assessment of steel frames. Engineering Structures, 33(9), 2535-2546.

[REF7] Vielma-Perez J.C., Porcu M.C., Gomez-Fuentes M.A. (2020), Non-linear analyses to assess the seismic performance of RC buildings retrofitted with FRP, Revista Internacional de Métodos Numéricos para Cálculo y Diseño en Ingeniería, vol. 36, (2), 24

Author Response

Please see the attached PDF file.

Reviewer 2 Report

This paper presented the extensive research on the prediction of the maximum and cumulative responses of a fourteen-storey reinforced concrete base-isolated building using pushover analyses, including the maximum and cumulative responses of the first and higher modes, the maximum local responses and the cumulative strain energies of lead-rubber bearings and steel dampers. The numerical results are genuine, initiative and practically useful. The conclusions are inclusive, sound and convincing. The paper is very well prepared, edited and written, including tables, figures, equations and references. The paper is an excellent technical paper and provides much new information for practical engineers and numerical analysts so it is worthwhile to publish. There are only some small editorial and grammatical issues, or typos, which are clearly marked in the paper and need to be considered and revised before the paper can be published, but these issues are regarded as minor ones.

Round 2

Reviewer 1 Report

The paper has been improved accordingly to the reviewer's comments, although the authors did not agree with the reviewer's suggestions for bibliography addictions.

The paper can be published in the present form.